

# Correlation of ankle dorsiflexion range of motion with lower-limb kinetic chain function and hop test performance in healthy male recreational athletes

Haifa Saleh Almansoof, Shibili Nuhmani and Qassim Muaidi

Department of Physical Therapy, College of Applied Medical Sciences, Imam Abdulrahman Bin Faisal University, Dammam, Eastern Province, Saudi Arabia

## ABSTRACT

**Background**. The study aims to identify the correlation of ankle dorsiflexion range-of-motion (ADROM) (with its related gastrocnemius and soleus extensibility) with lower-limb kinetic chain function and hop test performance in young healthy recreational athletes.

**Methods**. Twenty-one young male healthy recreational athletes were tested for ADROM, gastrocnemius and soleus extensibility, lower-limb kinetic chain function with the closed kinetic chain lower extremity stability test (CKCLEST) and hop test performance with the single-leg hop for distance test (SHDT) and side hop test (SHT).

**Results**. There was a positive significant (rho = 0.514, 95% CI [0.092–0.779], $P < 0.01$) correlation between the dominant lower-limb weight-bearing/closed-chain ADROM (that represented the soleus extensibility) and the CKCLEST. There were no significant correlations between the study performance-based tests and open-chain ADROM ($P > 0.05$).

**Conclusion**. The CKCLEST is positively and significantly correlated with SHT and weight-bearing ADROM with knee flexion (and its related soleus extensibility) which suggests comparability among them. Open-chain ADROM has a negligible and non-significant correlation with the readings of this study performance-based tests suggesting that it is probably not an essential construct of their execution. To the best of our knowledge, this study is the first to investigate these correlations.

## INTRODUCTION

The kinetic chain is the limb muscle coordinated sequential activations to execute the activity, which demands optimal flexibility, strength, proprioception, and endurance of the body parts (*Sciascia & Cromwell, 2012*). Closed kinetic chain (CKC) activity, where the limb terminal part meets a considerable resistance and is not free to move, generates forces that decrease joint shear, boosts agonist-antagonist muscle coactivation (*Lutz et al., 1993*), and simulates weight-bearing activities such as jumping and running (*Kibler, 2000*). In sports, the kinetic chain means the biomechanical system used by the athlete

Corresponding author
Shibili Nuhmani,
snuhmani@iau.edu.sa

to perform the sport-specific tasks (*Kibler, 2014*). Alteration or breakage in the kinetic chain can be due to many factors, including decreased muscle extensibility and joint range-of-motion (ROM) (*Kibler, 2014*). It has been theorized that a joint motion deficit can cause abnormal movement series and alteration (compensation) in muscle activity in the adjacent segments while performing a CKC (*Alizadeh, Alizadeh & Rajabi, 2017*). As a result, new motor programs/patterns can develop, which may cause overuse/overload injuries (*Alizadeh, Alizadeh & Rajabi, 2017*). Proper kinetic chain function can lessen the risk of injury, as proven in a study done on university-aged recreational female athletes (*Cannon, Cambridge & McGill, 2019*). Those with less dynamic valgus during drop-jumping activity had a greater lumbar spine and gluteal elastic resistance to rotational joint motion (*Cannon, Cambridge & McGill, 2019*). Consequently, they had less risk of sustaining anterior cruciate ligament injury (ACLI) (*Cannon, Cambridge & McGill, 2019*). The CKC function can be tested quantitatively using the CKC lower extremity stability test (CKCLEST), a valid and reliable non-plyometric performance test (*Arikan et al., 2021*). In short, the kinetic chain concept implies that the impairment of a joint can precipitate injuries to other joints (*Pattyn et al., 2011*). Moreover, efficient energy transfers along the kinetic chain were linked to a lower risk of injury and higher performance (*Augustus et al., 2021*).

The hop test is a performance-based test that echoes the combined influence of neuromuscular control, capacity to generate power, speed, acceleration, deceleration, rebound, direction changeability, and trust in the limb joints throughout its propulsive and landing phases (*Kotsifaki et al., 2021*; *Manske & Reiman, 2013*; *Petschnig, Baron & Albrecht, 1998*). The single-leg hop for distance test (SHDT) is valid and reliable in evaluating muscle strength/power deficits (*Reid et al., 2007*). The SHDT can detect the risk of time-loss risk lower-limb and low-back injuries in collegiate athletes throughout the preseason (*Brumitt et al., 2013*). Furthermore, it can distinguish between normal and unstable ankles to a moderate extent (*Vogler et al., 2017*). The side hop test (SHT) is a valid and reliable test for evaluating strength under the state of fatigue *via* controlled, fast, and repetitive lateral hops (*Kockum & Heijne, 2015*). The SHT represents the overall ankle readiness to practice sports activities (*Greisberg et al., 2019*).

Ankle dorsiflexion ROM (ADROM) assessment in the clinical setting is crucial as it may be linked to lower-limbs harmful movement patterns (*Lima et al., 2018*). Moreover, ADROM asymmetry was among the highly sensitive and specific risk factors of time-loss injury in military service members (*Teyhen et al., 2020*). Decreased ADROM has been linked to ankle, ACL, tendon (Achilles/patellar), and hamstring injuries (*Moreno-Pérez et al., 2020*). Furthermore, decreased ADROM is a significant shoulder/elbow injury risk factor (*Shitara et al., 2021*). Increased ADROM may maintain or improve dynamic postural control and prevent injury occurrence (*Burns et al., 2017*; *Nakagawa & Petersen, 2018*). An increase in ADROM by 1° can reduce the ACLI probability by 62 percentage in male athletes (*Amraee et al., 2017*). A study on young baseball players found that if ADROM increases by 3.6°, the upper-limb injury risk reduces by 19%, which is a considerable effect (*Shitara et al., 2021*).

ADROM is an indicator and clinical measure of landing quality and safety (*Malloy et al., 2015*). Decreased ADROM is a predictor of poor landing kinematics and kinetics

throughout the drop-jump activity in female athletes (*Malloy et al., 2015*). Decreased ADROM was linked with a significant reduction in sagittal plane joint displacement in the lower-limb joints in landing performance (*Howe et al., 2019*), which can raise the risk of knee injury (*Malloy et al., 2015*). While walking, decreased ADROM causes the pressures of body-weight-bearing to be shifted to the forefoot, leading to numerous lower-limb pathologies (*Abdulmassih et al., 2013*). Furthermore, decreased weight-bearing ADROM correlates significantly with the dynamic postural control tested by the Lower Quarter Y-Balance Test (YBT-LQ) (*Kang et al., 2015*) and Star Excursion Balance Test (SEBT) (*Burns et al., 2017*). ADROM under non-weight-bearing conditions was significantly correlated with balance as measured with the Neurocom Smart Balance Master's limits-of-stability test on spatiotemporal gait parameters using the GAITRite electronic walkway in young and healthy participants (*Norris et al., 2016*). ADROM was positively and significantly correlated with jumping performance in the Counter Movement Jump test in young, healthy soccer athletes (*Godinho et al., 2020*). Passive ADROM was significantly correlated with the quality of movement assessed with the lateral step-down test in physically active individuals (*Rabin et al., 2014*) and those with chronic ankle instability (*Grindstaff, Dolan & Morton, 2017*).

Injuries in sports practice are repeatedly linked with improper joint alignment and movement patterns (*Bates et al., 2015*). Acute injuries in sports mainly affect athletes during performing power events like sprinting, sharp direction changes, jumping, and landing (*Bates et al., 2015*). Knowing risk factors is important but being conscious of how they are interlinked and interacting is crucial for effective injury rehabilitation and prevention (*Bittencourt et al., 2016*). To the best of our knowledge, no previous research has studied the correlation of the ADROM with lower-limb kinetic chain function and hop test performance in young, healthy athletes. The study aims to identify the correlation of ADROM (with its related gastrocnemius and soleus extensibility) with lower-limb kinetic chain function (using the CKCLEST) and hop test performance (using the SHDT and SHT) in young, healthy, and recreational athletes. Such investigation could add to the body of knowledge of sports performance, screening, injury incidence/recurrence prevention, and rehabilitation in young, healthy athletes. Consequently, this can help decrease the medical cost and time lost from sports participation/competition.

## METHODS

### Participants

The current study is a cross-sectional study conducted at the Physio-Trio physical therapy clinic in Riyadh city in Saudi Arabia. A power analysis was used (https://www.ai-therapy.com/psychology-statistics/sample-size-calculator) to calculate the sample size. The calculations used data from a previous study (*Swearingen et al., 2011*) that investigated the correlation between the hop tests and the single-leg vertical jumping in healthy males and females between 18 and 30 years old. The following values were utilized to calculate the sample size in this study: correlation coefficient of 0.7, significance level alpha value of 0.05, and statistical power of 0.8. The calculation gave a sample size of 21 participants.

**Table 1** Demographic characteristics of the study participants.

| Variable | Mean ± SD |
| --- | --- |
| Age (in years) | 21.52 ± 3.03 |
| Height (centimeters) | 173.29 ± 6.57 |
| Mass (kilograms) | 64.40 ± 7.17 |
| Body mass index (BMI) (kilograms per meter square) | 21.51 ± 1.95 |
| Dominant calf circumference (centimeters) | 35.67 ± 2.04 |
| Non dominant calf circumference (centimeters) | 35.52 ± 1.97 |
| Main sports practice (years) | 8.90 ± 5.15 |
| Main sports practice (days/week) | 5.10 ± 1.58 |
| Main sports practice (minutes/day) | 137.62 ± 61.07 |

Notes.
SD, Standard Deviation; BMI, Body Mass Index.

Twenty-one young (18–35 years old) male recreational athletes (who regularly practice one to three times per week in sports events (*Chappell et al., 2002*)) with a mean age of 21.52 ±3.03 years, a height of 173.29 ±6.57 cm, weight of 64.40 ±7.17 kg, body mass index (BMI) of 21.51 ±1.95 kg/m$^2$, participated in the study. The demographic characteristics of the participants are available in Table 1. Athletes with a history of back/lower-limb injury that required medical attention in the past six months, trunk/lower-limb surgery, medical condition (chronic or systemic), deep venous thrombosis in the lower limbs, concussion, smoking, or red flags (*i.e.,* night sweats, and unexplained weight loss) and athletes who took medications (which may affect the execution of the tests) or had any biomechanical abnormalities, deformity, neurological symptoms which may affect the testing, physical pain, ankle instability, and hamstring tightness were excluded from the study. Athletes with BMI above or below normal (normal BMI is > 18.5 to < 24.9 kg/m$^2$) were excluded. Additionally, athletes who reported having mental health conditions, were on psychiatric medication, or disturbed sleeping were also excluded from the study. The participants signed the written informed consent statement form prior to participation in the study. Ten participants practiced more than one type of sport. The included participants' main sports that were practiced regularly were soccer (11 participants), karate (four participants), basketball (two participants), running (one participant), paddle tennis (one participant), archery and running (one participant), and archery and calisthenics (one participant).

Data collection was started from December 2021 to February 2022 (inclusive) and took several forms. An invitation through advertisement using social media. Sports coaches in fitness centers were given the advertisement to spread among their customers. There were 66 respondents, 21 were included, and 45 were excluded. The excluded respondents were 45 (34 males and 11 females). Those excluded were due to being below the required age (four athletes), above the required age (four athletes), with high BMI (five athletes), smokers (three athletes), injured (13 athletes), have physical pain (three athletes), have surgical history, history of head injury (one athlete), suffer chronic medical conditions (three athletes) and one respondent who was not an athlete. The institutional review board (IRB) ethical approval (IRB –PGS –2021 –03 –475) was earned from Imam Abdulrahman Bin Faisal University on the 19th of December 2021.

## Procedure

Before testing the study variables, the principal investigator described the testing procedure to the study participants. The data collection process was done by getting the demographical data, screening for eligibility, getting the signed informed consent form from the participant, conducting the ADROM measurement, and performance-based tests. The data collection sheet was designed to document the data collected from the testing procedure. The variables were ADROM, gastrocnemius-soleus extensibility (represented by the weight-bearing ADROM), lower-limb kinetic chain function, and single-leg hop performance. The study variables were tested with outcome measures described in Table 2 with the testing process timeline. The principal investigator conducted all the testing to reduce the measurement variability (*Encarnación-Martínez et al., 2020*). The dominant lower-limb dominancy was recognized by asking the athlete to execute three tests that the athlete must instinctively (without being aware of the purpose) choose one lower limb to do at least two ball kicking, fake fire extinguishing, and drawing figures on the ground tasks. The chosen lower limb for the minimum of two of the above-mentioned tasks was considered the dominant lower limb (*Schneiders et al., 2010*; *Vaisman et al., 2017*). The evening time was the period of the day that the tests were performed for all the study participants. The testing of the lower-limb kinetic chain function, and single-leg hop performance was started with warming up session with five minutes lower-limbs cycling with stationary bike at a self-selected and self-perceived moderate level of intensity (*Negrete et al., 2021*) to increase muscle temperature (*Bishop, 2003*). The order of the tests was random among the study participants to help in avoiding the possibility of that one test might affect the performance of the other. In the hop tests, a coin-flip governs which lower limb will be examined first (*Brumitt et al., 2013*) and both lower limbs were tested alternatively.

### Open-chain ADROM

The Modified Root protocol was used to evaluate the passive ADROM for both lower limbs (*Rowlett et al., 2019*). A universal goniometer (Elite Medical Instruments, Fullerton, CA, USA) was the tool to evaluate open-chain/non-weight-bearing ADROM in knee-angle-based positions: with the knee in 0-degree extension (position 1) and with the knee in 90 degrees flexion (position 2) (*Rowlett et al., 2019*) as illustrated in Fig. 1. The goniometer axis was put on the lateral malleolus. One goniometer arm was aligned with the fibula, indicating toward the fibular head. The other arm was put parallel to the fifth metatarsal longitudinal axis. The body landmarks were marked to have an accurate measurement (*Weaver, 2001*; *Fong et al., 2011*; *Krause et al., 2011*; *Rowlett et al., 2019*). One tester (the principal investigator) completed the joint ROM measurement because the assessment of the joint ROM with a universal goniometer has been shown to have greater intra-tester than inter-tester reliability (*Boone et al., 1978*).

### Gastrocnemius and soleus extensibility (weight-bearing ADROM with knee flexion/extension)

Muscle extensibility was measured by the universal goniometer (illustrated in Fig. 2) using the weight-bearing/closed-chain ADROM. The soleus extensibility was measured using the

**Table 2  The time-line of the process of testing the study variables.**

| Testing variables | Testing order |
| --- | --- |
| **First:**<br>**Using universal goniometer for ADROM testing:** | – Explaining testing procedure using the go-niometer to the participant and that he needs to be in prone position. |
| – With knee extension | – Then, testing both lower limbs for three trials. |
| – With knee flexion | |
| Average score was used for data analysis. | |
| **Second:**<br>**Using universal goniometer for muscle extensibility testing:** | – Explaining testing procedure using the go-niometer to the participant and that he needs to be standing with the tested lower limb flexed and then extended at the knee to the maximum before the heel got lifted from the floor. |
| – Weight-bearing ADROM with the knee extended (for the gastrocne-mius extensibility). | |
| – Weight-bearing ADROM with the knee flexed (for the soleus extensi-bility). | – Then, testing both lower limbs for three trials. |
| Average score is used for data analysis. | – Next, each test of the study performance-based tests in the third step got explained with the participant demonstrated the test to proof his understanding of the test before the scoring got started. |
| **Third:**<br>**The following performance-based tests are tested in a random order:**<br>– **CKCLEST** (for three trials with resting for one minute between trials).<br>– **Hop tests:**<br> – SHDT (for three trials).<br> – SHT (for three trials).<br>The best score among the three trials is the score used for data analysis. | Starting the third step with warming up session with five minutes lower-limbs cycling with sta-tionary bike at a self-selected and self-perceived moderate level of intensity (*Negrete et al., 2021*) to increase muscle temperature (*Bishop, 2003*). The order of the tests in the third step was ran-dom among the study participants to help in avoiding the possibility of that one test might af-fect the performance of the other.<br><br>In the hop tests, a coin-flip governs which lower limb will be examined first (*Brumitt et al., 2013*). Both lower limbs were tested alternatively. |

Notes.
ADROM, Ankle Dorsiflexion Range-of-Motion; CKCLEST, Closed Kinetic Chain Lower Extremity Stability Test; SHDT, Single-leg Hop for Distance Test; SHT, Side Hop Test.

weight-bearing ADROM with knee flexion. The gastrocnemius extensibility was measured using the weight-bearing ADROM with knee extension (*Encarnación-Martínez et al., 2020*).

Similar anatomical landmarks and universal goniometer arms placement were used for open-chain ADROM and extensibility (for gastrocnemius and soleus) testing. The universal goniometer stationary arm was placed on a marked fibular distal long axis (a line linking the center of the lateral malleolus with the center of the fibula head) (*Worrell, McCullough & Pfeiffer, 1994*). While the moveable arm of the universal goniometer was placed parallel to the fifth metatarsal for the open-chain measurements, it was placed parallel to the floor for the weight-bearing measurements (*Worrell, McCullough & Pfeiffer,*

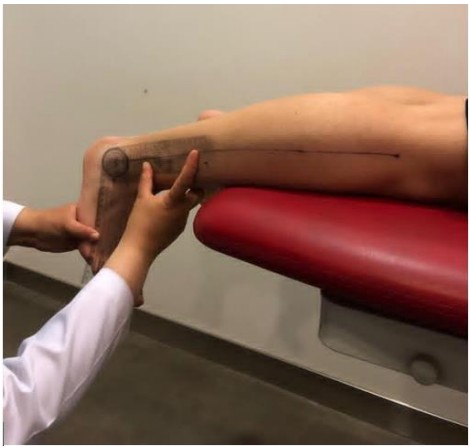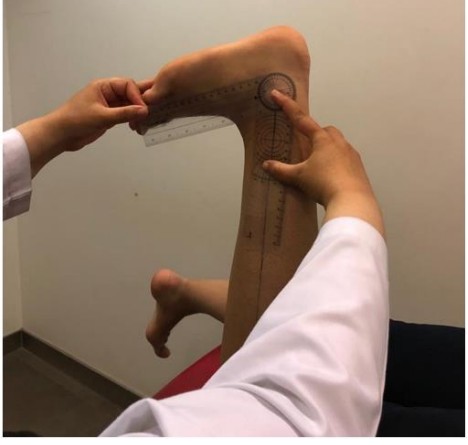

**Figure 1** **Modified Root position 1 (illustrated in the photo on the left) and position 2 (illustrated in the photo on the right) for open-chain ADROM testing.** The fibular distal long axis was marked to guide the placement of the universal goniometer stationary arm. The moveable arm was placed parallel to the fifth metatarsal.

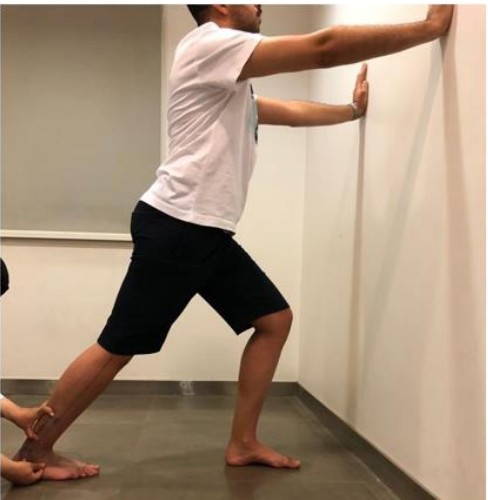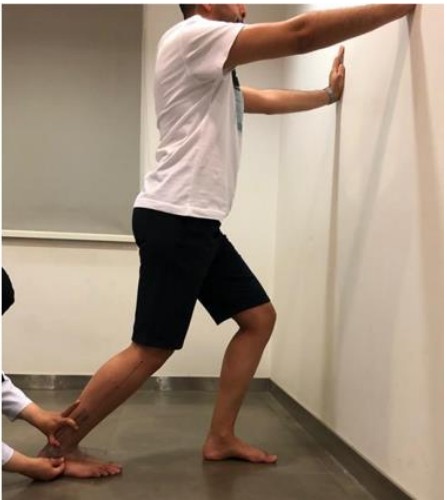

**Figure 2** **The gastrocnemius (illustrated in the photo on the left) and soleus (illustrated in the photo on the right) extensibility testing.** The universal goniometer stationary arm was placed on a marked fibular distal long axis. The moveable arm was placed parallel to the floor.

*1994*). The ADROM was recorded from each participant using the methods mentioned above during three trials, and the average ROM reading was calculated (*Rowlett et al., 2019*).

*Lower-limb performance tests*
*CKC lower-extremity stability test (CKCLEST)*
A stable base floor, mat, and stopwatch were the tools used to perform CKCLEST. The starting position of the CKCLEST was the plank position using the support of the forearms with the feet shoulder-width apart, the toes were in contact with the ground, and the body was in a straight line. While maintaining this initial body position, the athlete was asked to cross the outer side of the other foot with one foot and return. Then, the athlete made the same cross-and-return movement using the other foot. The athlete did this movement with both feet alternatively as fast as possible, as illustrated in Fig. 3. The number of repetitions in 15 s was recorded by counting the number of times the foot touched the other foot's lateral side and the floor. Three repetitions of the 15-second CKCLEST were performed with an interval of one minute between each test. Each athlete was allowed to perform the test once as a trial to be familiar before the actual test. Three trials were done, and the best trial score was used in the data analyses (*Arikan et al., 2021*).

*Single leg hop for distance test (SHDT)*
The athlete stood on the test lower limb and then hopped as far as he could and landed on the same lower limb with no extra hops until two-to-three seconds, as illustrated in Fig. 4. Then the investigator measured the distance (in centimeters using a tape measure) from the big toe at the push-off to the heel at the landing (*Gustavsson et al., 2006*). Single leg hop for distance test was accomplished with arms behind the back because it is the most sensitive procedure to identify poor functional performance (*Ageberg & Cronström, 2018*). A coin-flip governed which lower limb the athlete hopped with first (*Brumitt et al., 2013*). The single-legged hop test was executed barefoot (*Harrison et al., 2017*). The test was performed three times alternatively between the right and left lower limb (*Ageberg & Cronström, 2018*). The best distance achieved was the score needed for data analysis.

*Side hop test (SHT)*
The athlete stood on the tested leg, with hands behind the athlete's back, and jumped from side to side between two parallel tape strips (with a 40-centimetre distance between them), as illustrated in Fig. 5. The athlete was directed to jump as many times as he could throughout 30 s. The successful jumps repetition (without touching the tape) was documented (*Gustavsson et al., 2006*). The test was performed for three trials alternating the trials between lower limbs. The test was executed barefoot (*Harrison et al., 2017*). The best score (the one with the highest repetition) was the score needed for data analysis.

## Statistical analysis
The analysis was performed using the IBM SPSS version 27. Shapiro–Wilk test was used to check normal distribution ($P > 0.05$ was considered normal). The outcome data were presented as mean and standard deviation for the parametric data with a normal distribution. The outcome data were presented as median (25–75 percentile) for the variables that were not normally distributed and those that were nonparametric. Demographical analysis was used to summarize the athletes' age (in years), participation

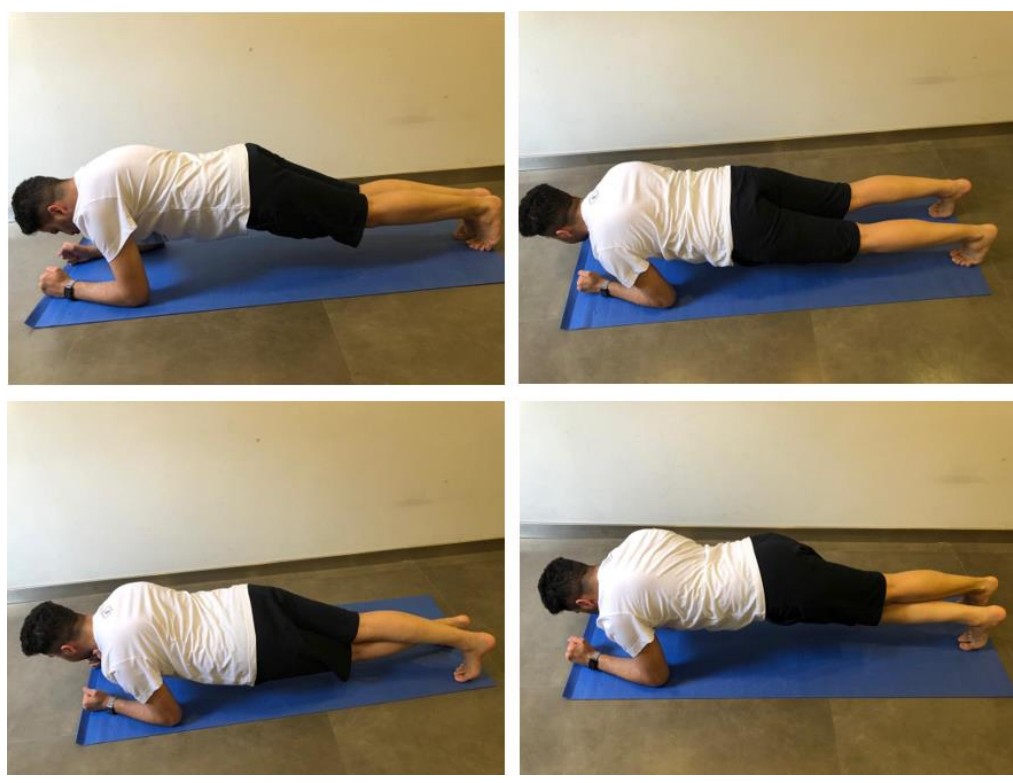

**Figure 3 CKCLEST test application.** The photos on the top demonstrate the starting position (the plank position using the support of the forearms with the feet shoulder-width apart, the toes in contact with the ground, and the body in a straight line). The photos on the bottom demonstrate the test application (alternative cross-and-return movement between lower limbs). The score is the cross-and-return movement repetition in 15 s.

in sport (in minutes per day, days per week and years), athlete mass (in kilograms), height (in centimeters), and calf muscle girth/maximal circumference (in centimeters).

The Pearson product-moment test was used to test the correlations between ADROM readings (in degrees) and single-leg hop distances (in centimeters) when the normal distribution was confirmed. Because the normal distribution was not confirmed, Spearman's rank test was used for correlating the dominant lower-limb open-chain ADROM (in degrees) with knee extension with other variables. Spearman's rank test was used to test the correlation between the ADROM (in degrees) and the SHT number of successful side-hops (in 30 s). Spearman's rank test was used to test the correlation between the ADROM (in degrees) and the CKCLEST number of cross-and-return movements (in 15 s). The Spearman's and Pearson's correlation coefficients interpretations to determine the strength of the correlations are negligible, weak, moderate, strong, and very strong for +/- (0.00 –0.10), (0.10 –0.39), (0.40 –0.69), (0.70 –0.89), and (0.90 –1.00), respectively (*Schober, Boer & Schwarte, 2018*). The correlation testing was of two-tailed directionality (*Cho & Abe, 2013*). For all correlations, the 95% confidence intervals (CI) were calculated. An alpha level ($P < 0.05$) was used for statistical significance (*Andrade, 2019*).

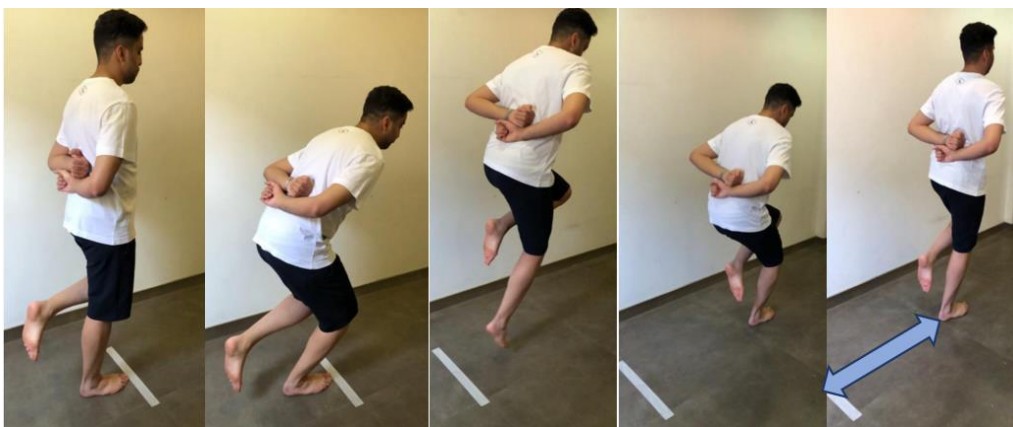

**Figure 4** **SHDT with bare feet and arms behind the back.** The score is the distance travelled in centimeters (represented by the blue double headed arrow) from the starting line (the athlete's big toe was behind the line) to the heel after landing.

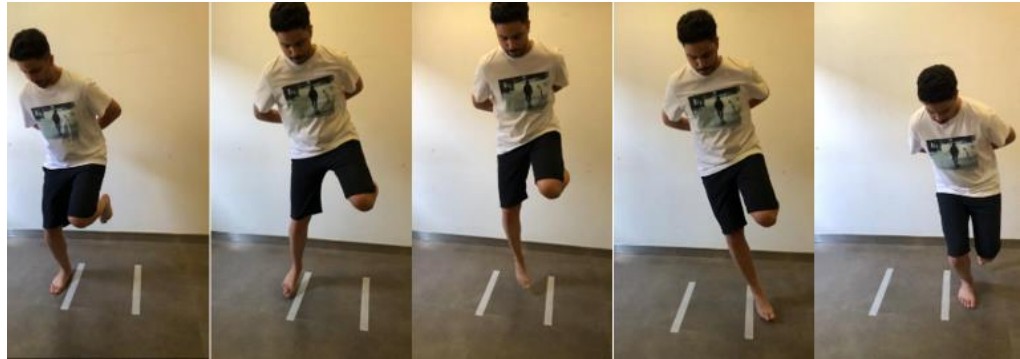

**Figure 5** **Illustration of the SHT.** The athlete was directed to jump as many times as he could throughout 30 s. The successful jumps repetition (without touching the tape on the floor) was the score needed.

## RESULTS

There was no significant difference between the current study participants in height, weight, BMI, maximal calf circumference, main sport practice years, or minutes of practicing the main sport per day. Nevertheless, there was a significant difference in age and the number of days of practicing the main sport per week. The normality test (Shapiro–Wilk test) for the study variables displayed that all the study variables were normally distributed except the dominant lower-limb open-chain ADROM with knee extension and CKCLEST (repetition/15 s) data. Table 3 displays the mean +/- standard deviation for each normally distributed study variable. Table 4 displays the median (25–75 percentile) for the dominant lower-limb ADROM with knee extension (since they were not normally distributed), CKCLEST, SHT (repetition/30 s) for the dominant lower-limb, and SHT (repetition/30 s) for the non-dominant lower-limb data.

**Table 3  Tested variables mean and standard deviation.**

| The normally distributed and parametric tested variables | Mean ±SD |
|---|---|
| Dominant lower-limb open-chain ADROM with knee flexion (degrees) | 25.62 ± 10.11 |
| Non-dominant lower-limb open-chain ADROM with knee flexion (degrees) | 25.14 ± 12.34 |
| Non-dominant lower-limb open-chain ADROM with knee extension (degrees) | 11.90 ± 7.96 |
| Dominant leg gastrocnemius extensibility (weight-bearing ADROM with knee extension) (degrees) | 29.90 ± 8.26 |
| Non-dominant lower-limb gastrocnemius extensibility (weight-bearing ADROM with knee extension) (degrees) | 29.33 ± 9.06 |
| Dominant lower-limb soleus extensibility (weight-bearing ADROM with knee flexion) (degrees) | 34.95 ± 9.05 |
| Non-dominant lower-limb soleus extensibility (weight-bearing ADROM with knee flexion) (degrees) | 34.67 ± 9.60 |
| SHDT (centimeters) for the dominant lower limb | 146.67 ± 25.10 |
| SHDT (centimeters) for the non-dominant lower limb | 147.67 ± 25.09 |

Notes.
ADROM, Ankle Dorsiflexion Range-of-Motion; SHDT, Single-leg Hop for Distance Test.

**Table 4  Tested variables' median and 25–75 percentiles.**

| The non-normally distributed/non-parametric tested variables | Median (25–75 percentiles) |
|---|---|
| Dominant lower-limb open-chain ADROM with knee extension (degrees) | 10 (6–13) |
| CKCLEST reading (repetition/15 s) | 22 (21–29) |
| SHT reading (repetition/30 s) for the dominant lower limb | 43 (34–53) |
| SHT reading (repetition/30 s) for the non-dominant lower limb | 44 (34.50–54.50) |

Notes.
ADROM, Ankle Dorsiflexion Range-of-Motion; CKCLEST, Closed Kinetic Chain Lower Extremity Stability Test; SHT, Side Hop Test.

Table 5 shows the 24 correlations tested among the study variables using Spearman for the non-parametric analysis and Pearson correlation tests for the parametric analysis. There was a positive, moderate (Spearman's correlation coefficient rho = 0.514, 95% CI [0.092–0.779]), and significant ($P < 0.01$) correlation between the dominant lower-limb weight-bearing/closed chain ADROM that represented the soleus muscle extensibility and the CKCLEST. This correlation was visually illustrated by the scatter plot in Fig. 6. Other correlations were non-significant and ranged between weak and negligible correlations, as displayed numerically in Table 5. In addition, a significant positive correlation between the CKCLEST and SHT was found in both dominant and non-dominant lower limbs (rho = 0.495, 95% CI [0.067–0.769], $P = 0.022$, and rho = 0.528, 95% [0.112–0.787], $P = 0.014$, respectively).

**Table 5  The correlations between ADROM and performance-based tests.**

| | Open-chain ADROM | | | | Weight-bearing ADROM | | | |
| --- | --- | --- | --- | --- | --- | --- | --- | --- |
| | Dominant lower limb | | Non-dominant lower limb | | Dominant lower limb | | Non-dominant lower limb | |
| | With knee flexion | With knee extension | With knee flexion | With knee extension | With knee flexion (Soleus extensibility) | With knee extension (Gastrocnemius extensibility) | With knee flexion (Soleus extensibility) | With knee extension (Gastrocnemius extensibility) |
| CKCLEST | rho = 0.255 95% CI [−0.211, .0627] $P$ = 0.264 | rho = 0.149 95% CI [−0.314, 0.555] $P$ = 0.518 | rho = 0.291 95% CI [−0.174, 0.650] $P$ = 0.200 | rho = 0.335 95% CI [−0.127, 0.677] $P$ = 0.138 | rho = 0.514* 95% CI [0.092, 0.779] $P$ = 0.017* | rho = 0.330 95% CI [−0.132, 0.674] $P$ = 0.145 | rho = 0.343 95% CI [−0.118, 0.682] $P$ = 0.128 | rho = 0.271 95% CI [−0.195, 0.637] $P$ = 0.234 |
| SHDT for the dominant lower limb | $r$ = −0.291 95% CI [−0.642, 0.161] $P$ = 0.201 | rho = 0.060 95% CI [−0.394, 0.489] $P$ = 0.798 | | | $r$ = 0.059 95% CI [−0.382, 0.479] $P$ = 0.799 | $r$ = 0.50 95% CI [−0.390, 0.471] $P$ = 0.830 | | |
| SHDT for the non-dominant lower limb | | | $r$ = −0.160 95% CI [−0.554, 0.292] $P$ = 0.488 | $r$ = −0.042 95% CI [−0.465, 0.397] $P$ = 0.856 | | | $r$ = −0.157 95% CI [−.551, 0.295] $P$ = 0.497 | $r$ = −0.239 95% CI [−0.608, 0.215] $P$ = 0.297 |
| SHT for the dominant lower limb | rho = 0.182 95% CI [−0.283, 0.578] $P$ = 0.429 | rho = 0. 288 95% CI [−0.177, 0.648] $P$ = 0.205 | | | rho = 0.198 95% CI [−0.268, 0.589] $P$ = 0.389 | rho = 0.145 95% CI [−0.318, 0.552] $P$ = 0.530 | | |
| SHT for the non-dominant lower limb | | | rho = 0.235 95% CI [−0.323, 0.614] $P$ = 0.305 | rho = 0.378 95% CI [−0.078, 0.703] $P$ = 0.091 | | | rho = 0.252 95% CI [−0.214, 0.625] $P$ = 0.270 | rho = 0.030 95% CI [−0.418, 0.466] $P$ = 0.898 |

**Notes.**

ADROM, Ankle Dorsiflexion Range-of-Motion; CKCLEST, Closed Kinetic Chain Lower Extremity Stability Test; SHDT, Single-leg Hop for Distance Test; SHT, Side Hop Test; CI, Confidence Interval; rho, Spearman's Correlation Coefficient; $r$, Pearson's Correlation coefficient; $P$, the probability value of getting the resulted data when the null hypothesis is assumed to be true.

*$P < 0.05$, the significance level was set at less than 0.05 with the star sign indicating the statistically significant finding.

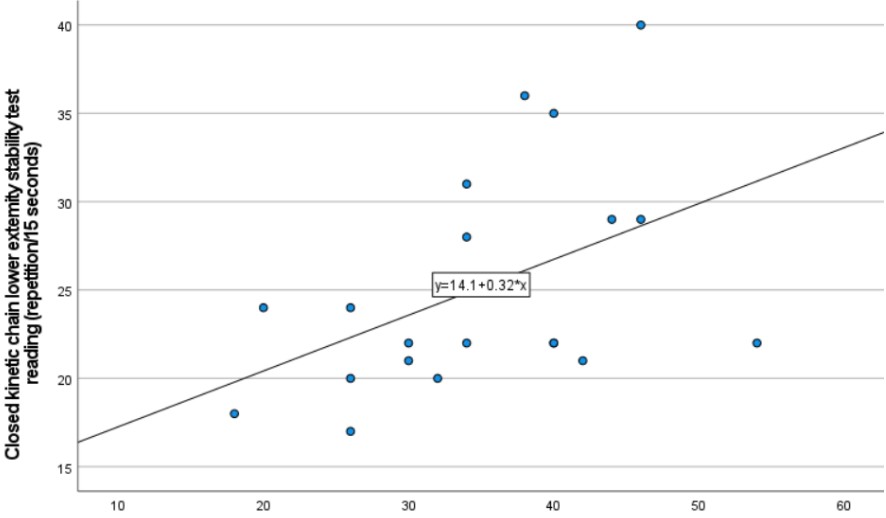

**Figure 6  Scatter plot illustrating visually the positive and moderately strong correlation between dominant lower-limb soleus extensibility represented by the weight-bearing ADROM with knee flexion and the CKCLEST.**

## DISCUSSION

The current study is the first (to the best of our knowledge) to investigate the correlation of the ADROM and its related gastrocnemius-soleus muscle extensibility with the hop test performance and the CKC function of the lower limbs. In young and healthy male recreational athletes, the current study found that the dominant lower-limb weight-bearing ADROM with knee flexion and its related soleus muscle extensibility was positively and significantly correlated with CKCLEST. Other correlations lack strength and significance.

Performing the CKCLEST requires repetitive mid-line crossing alternative reciprocal movements between lower limbs while maintaining the planking with both ankles in dorsiflexion for 15 s (*Arikan et al., 2021*). It could be that such a demanding and integrated task requires neurocognitive function (an integrated task that requires visual focus, self-monitoring, agility, dual tasking, accurate motor performance, reaction time, and speed (*Herman et al., 2015*)). The neurocognitive function may be associated with brain areas found in a previous study (*Francis et al., 2009*) to be activated with sustained ankle dorsiflexion. Furthermore, the ankle dorsiflexion could enhance the abdominal isometric action related deep musculofascial corset effect on the lumbopelvic region and its associated improved lumbar spinal stability (*Chon, Chang & You, 2010*) required to perform the CKCLEST test.

Moreover, the soleus muscle extensibility could acquire the link with the CKC function through being a part of the superficial back line (one of the myofascial chains) (*Myers, 2012*). The superficial back line can facilitate effective force transmission between the core and limbs by integrating the tension across it (*Krause et al., 2016*). Therefore, the higher soleus extensibility may help integrate the tension along the superficial back line to facilitate force transmission to execute the CKCLEST efficiently. A previous study (*Outram, Wilson & Wesselingh, 2020*) found strong, positive, and significant associations between the weight-bearing ADROM with knee flexion and backswing-related shoulder rotations, downswing-related pelvis and shoulder rotations, and pelvis, shoulder, and upper arm segments' velocities in male young healthy golfers. The CKCLEST requires pelvis rotations with speed to accomplish the lower-limbs' alternative mid-line crossing movements. Therefore, it can be derived that weight-bearing flexed-knee ADROM is linked to the diagonal/crosswise whole-body contribution in the CKCLEST performance.

There was a negligible and non-significant correlation between the non-weight-bearing open-chain ADROM and the CKCLEST. The closed-chain position augments the proprioceptive stimuli (*Kibler & Livingston, 2001*). These augmented proprioceptive stimuli could result in improved lower-limb activity coordination (*Tuthill & Azim, 2018*) needed in the CKCLEST execution. Therefore, it could be that the open-chain position lacked the action-specific alignment and its neuromuscular implications (augmented proprioceptive stimuli) where the ankle ROM could be relevant to the CKCLEST execution. The probable interpretation of the non-significant correlation between the single leg hop tests and the ADROM and its related soleus-gastrocnemius extensibility is likely to be joint-contribution related. The single-leg hop performance was correlated with hip function and

the lower-limb proximal joints and muscles involvement than the ankle joint and muscles (*Kotsifaki et al., 2021*; *Ono et al., 2021*).

## Additional findings
### The CKCLEST and SHT correlation
The correlation between the CKCLEST and SHT was positive and significant for the dominant and non-dominant lower limbs. To the best of our knowledge, this is the first study to find this correlation. The quick side hop performance was found to reflect hip function in terms of hip flexion, extension, and abduction torque in competitive sports performers (*Ono et al., 2021*). Moreover, hip abduction torque was a core stability booster while executing the single-leg side-stepping (*Inaba et al., 2013*). Therefore, there is a possibility that the SHT-related lower-limb biomechanical constructs mimic those of the CKCLEST.

The study presented preliminary data about the prevalence of decreased ADROM. Regarding the open-chain ADROM with knee extension, the study found that 17 out of the 21 participants (81%) have less than normal in dominant and non-dominant lower limbs (the normal ADROM with knee extension is 20 degrees (*Coetzee & Castro, 2004*)). As for compatibility with this findings, it was found (in a previous study) that young male soccer players may manifest a marked and early decrease in open-chain ADROM with knee extension from the first few years of practicing sports activities (*Francia et al., 2021*). For the open-chain ADROM with knee flexion, 15 out of 21 participant (71%) have decreased range in the dominant lower limb (the normal ADROM with knee extension is 35 degrees (*Baumbach et al., 2016*)). In the non-dominant lower limb, 17 out of the 21 participants (81%) were with decreased range.

In the study sample, the weight-bearing closed-chain ADROM with knee flexion (representing the soleus extensibility) was decreased in 16 out of 21 participants (76%) in the dominant lower limb and in 16 out of 21 participants (76%) had decreased range in the non-dominant lower limb. The normal weight-bearing closed-chain ADROM with knee flexion is 43 degrees (*Baumbach et al., 2016*). For the weight-bearing closed-chain ADROM with knee extension (representing the gastrocnemius extensibility), there were 20 out of 21 participants (95%) with a decreased range in the dominant lower limb. Likewise, 20 out of 21 participants had decreased range in the non-dominant lower limb. The normal weight-bearing closed-chain ADROM with knee extension is 34 degrees (*Baumbach et al., 2016*). Therefore, the study participants had a high prevalence of triceps surae decreased extensibility (in gastrocnemius more than soleus).

The possible explanation of the decreased ADROM to less than normal in the study participants can be made clear with the following correlations. In the study participants, a positive and significant (rho $= 0.504$, 95% CI [0.079–0.774], $P = 0.02$) correlation was found between the soleus extensibility and open-chain ADROM with knee extension of the dominant lower limb. Similarly, in the non-dominant lower limb, the correlation was positive and highly significant with soleus ($r = 0.705$, 95% CI [0.393–0.871], $P < 0.001$) and gastrocnemius extensibility ($r = 0.609$, 95% CI [0.240–0.824], $P = 0.003$). Likewise, in the dominant and non-dominant lower limbs, the correlation between the soleus

extensibility and open-chain ADROM with knee flexion was positive ($r = 0.702$, 95% CI [0.387–0.870], and 0.812, 95% CI [0.525–0.906], respectively) and highly significant ($P<0.001$). In the dominant and non-dominant lower limbs, the correlation between gastrocnemius extensibility and open-chain ADROM with knee flexion was positive ($r = 0.567$, 95% CI [0.179–0.802], and 0.805, 95% CI [0.572–0.918], respectively) and highly significant ($P = 0.007$ and $P < 0.001$, respectively). Moreover, it was found in a previous study (*Guillén-Rogel, San Emeterio & Marín, 2017*) that there was a significant positive link between the ADROM and ankle dorsiflexors strength in young and healthy male students. So, it can be derived that decreased ADROM is possibly linked to decreased plantar flexors' (soleus and gastrocnemius) extensibility and dorsiflexors' strength.

Knowing that the sports-related repetitive movements can shorten and thicken the fascia around the overused muscles (*De Witt & Venter, 2009*), affecting their extensibility. Therefore, the decreased soleus and gastrocnemius extensibility in the study sample can be referred to their extensive use in sports as confirmed in the following lines. In the running, just before the toe-off phase, the soleus and gastrocnemius absorb the peak mechanical power (*Sasaki & Neptune, 2006*). This peak power absorption increases when the ankle dorsiflexion is decreased (*Pope, Herbert & Kirwan, 1998*; *Kaufman et al., 1999*). Furthermore, in fast arm movements, the muscle activation sequence starts in the contralateral gastrocnemius (*Zattara & Bouisset, 1988*) and then through the core and thoracolumbar fascia reaches up to the arm (*Cordo & Nashner, 1982*; *Hirashima et al., 2002*). The ankle is a key joint to perform optimal vertical jumping (*Kilic et al., 2017*). In the jumping take-off phase, the mechanical energy flows through the ankle joint that is adapting its plantarflexion motion making 23% of the velocity (*Hubley & Wells, 1983*), which is dependent on the force amount generated by the gastrocnemius muscle (*Bobbert & van Zandwijk, 1999*). In landing from a jump, the gastrocnemius-soleus complex provides 44% of the energy absorption total muscle work (*Devita & Skelly, 1992*) and slows the ground reaction forces propagation speed to the knee (*Boden et al., 2010*).

On the other hand, in the study sample, it was found that 19 out of 21 participants (90%) had passive ankle plantarflexion ROM that is above the normal range in the dominant lower limb with mean of 66.3 degrees. Moreover, we found that all participants have beyond normal ankle plantarflexion ROM in their non-dominant lower limb, with a mean of 68.3 degrees. The normal plantarflexion passive ROM is 50.0 degrees (*Alazzawi et al., 2017*).

For the dominant lower limb, this finding can be justified when the study found a positive and highly significant correlation between the time spent practicing the main sport per day and passive ankle plantarflexion ROM in the dominant lower limb ($r = 0.677$, 95% CI [0.346–0.858], $P < 0.001$). This correlation was weak and non-significant ($r = 0.280$, 95% CI [−0.172–0.635], $P = 0.218$) in the non-dominant lower limb. The study participants' main sports involve many stretch-shortening activities (*e.g.*, running, sprinting, side hopping, and vertical jumping). Previous studies (*INal, Erbuğ & Kotzamanidis, 2012*; *Sugiyama et al., 2014*) confirmed the relationship between the dominant lower limb higher ankle plantarflexion ROM and the stretch-shortening activities' performance in young and healthy male athletes. A previous study (*INal, Erbuğ & Kotzamanidis, 2012*) found

that the 100-meter sprint running time was positively and significantly correlated with the ankle plantarflexion ROM in the dominant lower limb in young, healthy male sprinters. Likewise, it was found in collegiate young, healthy male basketball players during the running-single-leg-jumps that the ankle plantarflexion joint angles were significantly higher in the dominant than the non-dominant lower limb during the take-off phase (*Sugiyama et al., 2014*). It was found that the take-off phase higher ankle plantarflexion angles were linked with the effective ability to transfer the velocity of the run-up phase to higher jump height (*Sugiyama et al., 2014*). In side-hopping, at the end of the contact phase and preparation of the flight phase, the gastrocnemius-soleus muscle complex contract strongly to lift the body weight upwards (*Yoshida, Taniguchi & Katayose, 2011*). This strong contraction produces a fast ankle plantarflexion that has to be controlled by the tibialis anterior to properly adjust the ankle position (*Yoshida, Taniguchi & Katayose, 2011*).

Additionally, activities like kicking the ball that is repetitively practiced in football code sports demand more plantarflexion ROM and no dorsiflexion (*Líška et al., 2021*). Furthermore, it was found that young, healthy recreational athletes with decreased weight-bearing ankle dorsiflexion ROM with knee flexion (the mean was 36.3 degrees) tended to acquire greater ankle plantar flexion and knee extension at initial landing contact in young, healthy recreational athletes (*Howe et al., 2019*). Therefore, it can be derived that the increased ankle plantarflexion ROM in the study participants was possibly related to the time spent per day practicing stretch-shortening activities and decreased plantar flexors' (gastrocnemius and soleus) extensibility.

From the above mentioned discussed information about ADROM and ankle plantarflexion findings in the study sample, it can be derived that the possible explanation of the participants' increased range of ankle plantarflexion and decreased ADROM is linked to the sports-practice imposed musculoskeletal maladaptation and insufficient recovery after the training. The study is the first to have preliminary information about the prevalence of increased ankle plantarflexion ROM, decreased ankle dorsiflexion ROM, and decreased soleus and gastrocnemius muscle extensibility among healthy young recreational male athletes in Saudi Arabia. No such previous study was done in Saudi Arabia.

### The ankle ROM correlation between dominant and non-dominant lower limbs

In terms of ankle plantar flexion ROM, the study found that the dominant and non-dominant lower limbs are weakly and non-significantly correlated ($r = 0.398$, 95% CI [$-0.041$–$0.708$], $P = 0.074$). However, the ADROM was positively, strongly, and significantly correlated between the dominant and non-dominant lower limbs. In details, for open-chain ADROM with knee flexion ($r = 0.703$, 95% CI [$0.389$–$0.870$], $P < 0.001$) and with knee extension (rho = $0.606$, 95% CI [$0.223$–$0.827$], $P = 0.004$). For weight-bearing closed-chain ADROM with knee flexion ($r = 0.782$, 95% CI [$0.529$–$0.908$], $P < 0.001$) and knee extension ($r = 0.852$, 95% CI [$0.664$–$0.938$], $P < 0.001$). As for compatibility with such findings of the current study, a previous study found a positive and significant correlation between non-weight-bearing and weight-bearing ADROM with knee flexion on the dominant and non-dominant lower limbs (*Rabin et al., 2014*).

From a joint motion perspective, the ankle dorsiflexors and plantar flexors are crucial (*Lauber, Gollhofer & Taube, 2018*). However, during functional activities, the soleus (plantar flexor) is a high-force producer, while the tibialis anterior (dorsiflexor) generates lower forces (*Lieber & Fridén, 2000*). The training-related brain cortical adaptation is more optimal for the most proficient/dominant limb (*Farthing, 2009*). Moreover, previous studies' findings (*INal, Erbuğ & Kotzamanidis, 2012*; *Sugiyama et al., 2014*) proved the relationship between the dominant lower-limb higher ankle plantarflexion ROM and stretch-shortening activities' performance in young, healthy male athletes. So, since the dominant lower-limb plantar flexors are more proficient in producing force to help execute the activity, the ankle plantarflexion ROM may become uncorrelated between the dominant and the non-dominant lower limb.

## Study limitations

There are some limitations in the study. The study is a cross-sectional study. Therefore, no causal inferences can be concluded (*Rohrer, 2018*). There was a lack of sufficient control over the psychological condition and motivation to perform the tests. The psychological aspect that could affect the execution of the performance tests was not addressed in this study. Although, it is known that the athlete's motivation to practice sports increases when they perceive high self-confidence, self-perception, and self-trust in their abilities (*Sari et al., 2015*). However, in the study eligibility screening of the athletes, they were asked whether they had mental conditions (*e.g.*, depression, anxiety, sleeping disorders, or panic attacks). Though, those conditions were not ruled out by a specialized health care provider. The study participants were normal, young (18 to 27 years old), and recreational level male athletes. Therefore, there is a generalizability concern when using the current study's findings on non-athletes, females, inured athletes, or athletes with different health statuses, age groups, and practice levels. However, having no females in the current study included participants increased the sample homogeneity and, consequently, the study's internal validity (*Halperin, Pyne & Martin, 2015*). Furthermore, the study involved athletes from different sports, and many practiced more than one type of sports. This diversity in the sports practice could influence the current study's internal validity (*Halperin, Pyne & Martin, 2015*). However, such variety helped increase the external validity (*Ferguson, 2004*). Moreover, there was a lack of control over the nutritional status of the participants. However, the current study involved anthropometric data (the BMI and the calf maximal circumference) that were normally distributed and reflect elements of the nutritional status (*Bailey & Ferro-Luzzi, 1995*; *Bahat, 2021*).

## Recommendation for future studies

A deeper understanding of the link between ADROM and sports performance is important and can help refine the knowledge and efforts to provide better care to the athlete. So, it is recommended that future research addresses the current study research questions on another gender (*i.e.,* females), age groups, sports practice level (*i.e.,* elite), and health status (those with a health condition or injuries). Moreover, it is recommended to conduct the current research question on athletes who practice one type of sport to

explore the differences among the sports. Furthermore, since no causality inferences can be derived from correlational study results, conducting experimental studies is valuable to determine the cause-and-effect relationship. The study provided preliminary data concerning the deceased ADROM and increased plantarflexion ROM prevalence among healthy recreational male athletes in Saudi Arabia. The percentage in the study sample was high. Though, a prevalence study must be conducted to have a representative precise conclusion.

## Clinical implications

The current study findings suggested that, in young, healthy, and recreational male athletes, giving more attention to the dominant lower-limb weight-bearing ADROM with knee flexion and its related soleus extensibility in clinical examination and preseason screening could help add to the efforts of refining sports performance and minimizing the risk of injury. The CKCLEST and SHT scores are comparable. However, the CKCLEST puts compressive forces on the upper limbs, while the SHT puts compressive forces on the lower limbs. Therefore, the coach or health care provider can switch between CKCLEST and SHT to examine equivalent performance constructs based on the required or avoided compressive forces' effect on limbs. The high prevalence of ankle decreased dorsiflexion ROM, soleus-gastrocnemius extensibility, and increased plantar flexion among the study participants may encourage proactive efforts to screen athletes for these impairments, identify the vulnerable ones, provide them with proper intervention, and consequently improve sports performance and minimize injury risk. Regarding the ADROM, using the non-affected or contralateral lower limb as a reference in the clinical examination for young male recreational athletes is justified and reasonable. On the other hand, for the ankle plantar flexion ROM, the non-affected or contralateral lower limb should not be used as a clinical examination reference.

## CONCLUSION

According to the study principal findings attained in young, healthy, and recreational male athletes, weight-bearing/closed-chain ADROM with knee flexion and related soleus muscle extensibility in the dominant lower limb is positively and significantly correlated with CKC function in the lower limbs. Therefore, including the dominant lower-limb weight-bearing ADROM with knee flexion and its related soleus extensibility in the athletes' screening and clinical examination could add to the efforts and body of knowledge of sports performance and injury prevention. However, the non-weight-bearing/open-chain ADROM is not an essential component in the study performance-based tests (CKCLEST, SHDT, and SHT). Furthermore, the study's additional findings revealed a positive and significant correlation between the SHT and the CKCLEST, suggesting that they can provide comparable data on male, young, and healthy athletes. Thus, the SHT and the CKCLEST can be used interchangeably in clinical examination and screening. Moreover, the study is the first to report preliminary information about the prevalence of increased ankle plantarflexion ROM, decreased ankle dorsiflexion ROM, and decreased soleus and gastrocnemius muscle extensibility among young, healthy, and recreational male athletes in Saudi Arabia. This

preliminary data encourages conducting a prevalence study to provide representative information that can add to sports injury prevention efforts.

### Funding
The authors received no funding for this work.

### Competing Interests
Shibili Nuhmani is an Academic Editor for PeerJ.

### Author Contributions
- Haifa Saleh Almansoof conceived and designed the experiments, performed the experiments, prepared figures and/or tables, and approved the final draft.
- Shibili Nuhmani conceived and designed the experiments, analyzed the data, authored or reviewed drafts of the article, and approved the final draft.
- Qassim Muaidi analyzed the data, authored or reviewed drafts of the article, and approved the final draft.

### Data Availability
The raw data is available in the Supplemental Files.

### Supplemental Information
Supplemental information for this article can be found online at http://dx.doi.org/10.7717/peerj.14877#supplemental-information.

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
