# Peer review of "Correlation of ankle dorsiflexion range of motion with lower-limb kinetic chain function and hop test performance in healthy male recreational athletes"

_PeerJ, doi:10.7717/peerj.14877_

## Round 0.1 · original submission · Major Revisions

Although presenting some points of interest, the article requires major modifications. Please carefully consider the comments provided by the reviewers.

·

Basic reporting

The manuscript is clear and conform to standards. The English language is mainly correct but the punctuation and the citations in the text should be revised. Eg: line 33 (full stop missing), 50, 68-69; lines 60, 66, 72, 75, 78, 82, 89, 94, 162, 187,188, 192,194, 239, 246,247,257,274, 442,444.
The article structure fits the article purpose but it should be absolutly clear that the subjects in the figures authorized their publication or in the case they don't to cover their face so not to be identified.

line 224 - Please add: "....and the number of days.."

Experimental design

Title
As the sample only considers male recreational athletes, this aspect should be included in the title to clear it out.
Abstract
Is presented satisfactorily.
Introduction
The introduction is presented satisfactorily. However, it would be possible for the problem to be investigated to be better identified as well as the knowledge gap, and how the study contributes to filling that gap.

Methods
Some details of the methods used should be added. The test used to define the dominant leg definition must be clear for readers. Please also refer to the period of the day that the tests were performed and if all the athletes were tested in the same period of the day.
234 Please correct: “positive, highly significant (P<0.01), and moderate correlation (“
236 Please correct: “readings that represented the soleus muscle extensibility and the CKCLEST readings (figure 6).”

Validity of the findings

Results
Are presented satisfactorily.
Discussion
It should reaffirm the objectives and start discussing the results in the chronological order that appear in the item results. Please try to discuss the results clearly and based on objective results. Avoid generalizations impossible to do based on the sample used (eg: 421-423)
Please reconsider the limitations as they do not include all issues related to the sample (dimension and characteristics) and consider others that are not described as study aims (lines 433-434).
Conclusion
Is presented satisfactorily. However, it should include practical applications that the study's findings would indicate within the proposed problem and objective.
References
Please review the used references. A procedure can have different references but they should describe it and they should not be used separately to justify each step of the procedure, as it happens, for example in lines 163- 167.

Reviewer 2 ·

Basic reporting

The English language should be improved to ensure that an international audience can clearly understand your text. Some examples are highlighted along with suggestions so that the authors could improve. The word "current study" and "this study" is extended used. Please add appropriate changes to makes comprehension more easy.
Please consider reviewing your manuscript with a professional editing service or college that are proficient in English but also have a background in this subjects background.
The literature reference are adequate but with errors when cited or added to the text. Check the highlighted parts of the text.
The tables and figures are very illustrative but is laking a clear information of the appropriate place during the text, making difficult to understand the final version of the manuscript. See also the suggestions highlighted to improve. Add a clear information of the images that represents position 1, 2, 3 and 4 on figure 3.

Experimental design

The methods are well described, with sufficient detail and information, however is needed small changes. Please see the comments on the document.

Validity of the findings

no comment

Additional comments

The authors must carefully check the citations, giving special attention to the spaces between citations and the text, the citations of several authors and the correct format of the bibliography (se comments and highlighted text in the manuscript).
Your introduction is very detailed and informative, however I suggest that you improve some connections parts of the text to be more clearly understood.
The manuscript have some weakness in the discussion because of the repetitive of the information that is on the results section and the use of repetitive sentences and/or terms being difficult to clearly understood the main ideia.
The conclusion must be improved to better illustrate the principal findings.

Annotated reviews are not available for download in order to protect the identity of reviewers who chose to remain anonymous.

Reviewer 3 ·

Basic reporting

The manuscript “Correlation of ankle dorsiflexion range of motion with lower-limb kinetic chain function and hop test performance in healthy young athletes” has the main objective of identifying the correlation of ADROM (with its related gastrocnemius and soleus extensibility) with lower-limb kinetic chain function (using the CKCLEST) and hop test performance (using the SHDT and SHT) in young, healthy, and recreational athletes. This is an original and interesting manuscript, but I suggest some revisions so that the manuscript can be considered for publication in PeerJ.

Title of the manuscript: I suggest changing the word "young athletes" to "recreational athletes", I think it will be more appropriate.

Introduction:
The manuscript introduction is well written and can guide the reader to the problem and research objective.

Experimental design

Methodology:
I suggest that some clarifications be made by the authors so that the manuscript can be considered for publication.

I thank you for providing the raw data, however your supplemental files need more descriptive metadata identifiers to be useful to future readers. Although your results are compelling, the data analysis should be improved.

Line 122 - The raw data provided with the publication has 22 participants aged 18-27 years, this information is different from the information presented in the manuscript. In the manuscript, 21 participants aged 18-35 years are presented.
Line 123 - The study participants' mean and standard deviation data presented appear to be inconsistent with the raw data provided.

Line 135-137 - The raw data provided presents the data of a taekwondo athlete. This information does not seem to agree with the information described in the manuscript.

I suggest that all statistical analysis of the manuscript be reviewed against the raw data provided with the submitted manuscript. This new analysis may change the methodology, results, discussion and conclusions of the manuscript, so I suggest that the statistical review be done with great care.

Validity of the findings

Results: I suggest checking the results after the complete verification of the statistics referring to the raw data that was provided with the manuscript submission.

In table 5 it is important to present the confidence interval of the correlations.

Figure 6: Dear author, as you did not present in the methodology the intention to develop a linear regression, I did not understand the presentation of R2, could you make this information clearer? Does this result agree with the rho spearman used? Also, if you want to present this figure, I suggest using a more modern figure with a better-quality image.

Discussion: After reviewing the manuscript, I suggest ending the discussion with the practical application of the results of the study.

---

## Round 0.2 · Minor Revisions

Although the article has merit, it must consider the suggestions of reviewer 1.

·

Basic reporting

Please correct some English grammar and editing errors (eg line 86 The hop test; line 89 phases; 309 the quality; 370 were on psychiatric medication; 372 more than one type of sport; 414 do at least two ball-kicking; the lower limbs’; 712 presented preliminary).

Experimental design

Concerning the title, as previously stated, the sample in the study is only males even if the study itself, in the beginning, would include both genders, so that must be clear in the title.

Validity of the findings

Results
Additional findings should be presented in the results section.

Discussion
The subtitles “Preliminary information about decreased ADROM prevalence in Saudi Arabia:” and “Preliminary information about increased ankle plantar flexion ROM prevalence in Saudi Arabia:“ do not reflect the content of the paragraph. Prevalence in Saudi Arabia does not concern 21 participants and the study performed does not lead to that statement, which is recognized by the authors in lines 1157-1159 .

The sample size should be included in the limitations.

Reviewer 2 ·

Basic reporting

The English was substancial improved, however there is small changes needed in the text because is necessary the use of the past tense when stating results and talking about the research made.

Experimental design

The changes made clear the main research questions with the methods illustrating with sufficient detail and information.

Validity of the findings

Clinical implications must be presented after the conclusion.

Reviewer 3 ·

Basic reporting

no comment

Experimental design

no comment

Validity of the findings

no comment

Additional comments

Dear authors, I am happy to have collaborated with the development of the manuscript. Thank you for responding point-by-point to the requests I and the other reviewers made.

Best regards.

---

## Round 0.3 · Minor Revisions

Please consider the latest minor suggestions that can be observed in the attachment.

Reviewer 2 ·

Basic reporting

No comment

Experimental design

no comment

Validity of the findings

no comment

Additional comments

The authors have made substantial change allowing the paper to fulfill the principles to be published.

---

## Round 0.4 · accepted · Accept

The article was improved after the suggestions performed by the editor. The article can be accepted.